# Focused deterrence: A protocol for a realist multisite randomised controlled trial for evaluating a violence prevention intervention in the UK

**Tia Simanovic**[1], **Paul McFarlane**[2]*, **Iain Brennan**[1], **William Graham**[3], **Alex Sutherland**[4]

1 School of Criminology, Sociology and Policing, University of Hull, Hull, United Kingdom, 2 Department of Security and Crime Science, University College London, London, United Kingdom, 3 School of Business, Law and Social Science, Abertay University, Dundee, Scotland, United Kingdom, 4 University of Oxford, Oxford, United Kingdom

☯ These authors contributed equally to this work.
* p.mcfarlane@ucl.ac.uk

**Data Availability Statement:** No datasets were generated or analysed during the current study. All

## Abstract

### Introduction

Focused deterrence (FD) is a frequently cited intervention for preventing violence, particularly against violent urban gangs. The Youth Endowment Fund (YEF) believes it could be effective in the UK, based primarily on research conducted in the US. However, we contend that these studies have inadequate methodological designs, lack of rigorous testing, and small sample sizes. Therefore, the evidence supporting focused deterrence as an effective method, particularly outside the US, is inconclusive. The aim of the protocol is to better understand the potential effects of FD in the context of the UK, using a multisite evaluation experimental design to more closely investigate the evidence of its likely impact.

### Methods

We planned a realist randomised controlled trial. The design is focused on a multisite trial consisting of two-arm randomised experiments in five locations. Each trial location will test their implementation of a core programme specified by the funder. The multisite nature will allow us to understand differential impacts between locations, improving the external validity of the results. Participants will be randomly selected from a wider pool of eligible individuals for the intervention. We estimate a sample size of approximately $N = 1,700$ individuals is required. Based on this pooled sample size, a relative reduction of 26% would be detectable in 80% of trials. The trial is coupled with a formative process evaluation of delivery and fidelity. The formative evaluation will use a mixed methods design. The qualitative aspect will include semi-structured cross-sectional and longitudinal interviews with programme leads, programme delivery team, and programme participants, as well as observations of the meetings between the programme delivery team (i.e., community navigators/mentors) and programme participants. The quantitative data for the formative evaluation will be gathered

relevant data from this study will be made available upon study completion.

**Funding:** This work was supported by The Youth Endowment Fund under grant number 3928041.

**Competing interests:** The authors declare that no competing interests exist.

**Abbreviations:** CMO, Context Mechanism Outcome; FD, Focused Deterrence; ITT, Intention To Treat; PNC, Police National Computer; RCT, Randomised Control Trial; TiDieR, Template for Intervention Description and Replication; YEF, Youth Endowment Fund.

by the sites themselves and consist of routine outcome performance monitoring using administrative data. Sampling for interviews and observations will vary, with the researchers aiming for a higher number of individuals included in the first round of cross-sectional interviews and retaining as many as possible for repeat interviews and observations.

## Discussion

This protocol outlines the process and impact evaluation methodology for the most extensive multisite evaluation of focused deterrence to date in the UK. Spanning five distinct sites with seven trials, the evaluation includes a cohort of 2,000 individuals, marking it as the only multisite trial of focused deterrence. Employing an integrated realist evaluation framework, the study uses qualitative and quantitative research methods. The anticipated findings will offer pivotal insights for formulating future violence prevention policies in the UK. They are also expected to contribute significantly to the corpus of literature on violence prevention and intervention evaluation.

## Trial registration

**Protocol registration**: ISRCTN: 11650008 4th June 2023.

## Introduction

Violence is a leading cause of injury and mortality among young people worldwide [1]. It is associated with physical, psychological [2] and social harms [3] to victims, and potentially harmful contact with the criminal justice system for perpetrators [4, 5]. It also acts as a 'signal' of social disorder [6] to the wider population, affecting fear of crime and feelings of safety. Following a steady decline in all types of violence in the early 2000s, rates of police-recorded violence in the UK have trended upwards since 2013/14 [7]. In 2018, the Serious Violence Strategy [8] established "a new balance between prevention and effective law enforcement" (p.7) as the approach of the government of England and Wales to reducing violence, particularly violence involving young people. Since the publication of this strategy, over £800m has been allocated to finding and implementing effective violence prevention in England and Wales, often through the importation of interventions from the US. Although modest progress has been made in reducing violence in some areas [9], robust evidence of what works to reduce community violence in England and Wales remains elusive.

Focused deterrence is one of the most promising interventions for addressing community and youth violence [10, 11]. It has been implemented in dozens of jurisdictions across diverse urban settings, predominantly in the United States, with published evidence generally indicating positive effects in reducing serious violence [12, 13]. Similarities in the likely causes of community violence between the US and the UK mean that focused deterrence has the potential to be effective in the UK. However, past attempts to implement focused deterrence have met with implementation challenges [14] or inconclusive evidence of effectiveness [15, 16], and the importance of contextual differences in violence, policing, support services and community attitudes is not well understood. In particular, the prevalence of homicide and other serious violence in England and Wales—the rate of homicide is six times higher in the US [17] —means that there are likely to be fewer opportunities for intervention and rarer outcomes in England and Wales compared to the US.

Focused deterrence interventions are adapted to their location, but all include some combination of the following principles: (1) The intervention targets individuals and/or groups identified as being involved in serious violence in a community or area. These individuals are identified using intelligence from various official and community sources. They are informed that they are the focus of police attention and that various mechanisms will be activated, such as disruption to their daily activities and withdrawal of or reduction in statutory services, such as housing, should they continue to participate in violence. (2) They are also informed that, should they wish to desist from violence, a variety of support services will be made available to them to facilitate this desistance. (3) To underline the legitimacy and credibility of this message, it is conveyed by a combination of police, statutory and voluntary sector organisations, and community members in different formats. These three components—making violence less attractive through increased perceived probability of consequences, creating additional opportunities to desist from violence, and signalling community support for both components —bring together core theories of crime prevention that may have separate mechanisms but may also interact to affect violence outcomes.

Despite the promise of focused deterrence interventions, the empirical evidence support for its effectiveness is modest. Campbell systematic review of twenty-four studies [13] concluded that the intervention was associated with reduced serious violence but highlighted common methodological limitations of the literature. For example, none of the studies used a randomised allocation of treatment and control units, making them vulnerable to regression to the mean, and comparison groups were often other cities, or the same area at an earlier period, which may provide reliable counterfactuals. The risk of observed effects being influenced by regression to the mean is particularly acute in violence prevention interventions because they are often funded and implemented in response to an extreme level of violence in an area, which is inclined to return to less extreme levels. Most concerning is that many of these studies were statistically underpowered, arising from the use of groups/gangs or areas as the treatment unit and interventions being implemented in a single city or jurisdiction, meaning that any possible sample would be small. This statistical underpowering, combined with the other limitations, presents a heightened risk of Type 1 errors. In addition, the studies' outcomes, units of allocation and analysis, and conclusions varied, making for a mixed and inconsistent picture of effectiveness. To improve understanding of the effectiveness of focused deterrence, evaluations must implement more robust study designs using larger sample sizes.

## Aims

The study aims to evaluate the efficacy of focused deterrence as an intervention for reducing violent offending among individuals aged 14 and older who are at risk of violent offending in England. The study uses a multi-site parallel randomised controlled trial (RCT) design to assign participants to an intervention or a control group in a 1:1 ratio, with stratification based on age (adult/child status) and prior offending frequency. This design will facilitate a robust assessment of the intervention's impact on the occurrence of violent offending within this demographic.

## Methods and materials

### Realist multisite randomised controlled trial

To achieve the aims of the study, we designed it as a realist randomised controlled trial. The design includes a summative impact, multisite two-arm randomised evaluation, a formative process evaluation of both delivery and fidelity of the intervention, and longitudinal qualitative accounts of the intervention experience.

As one of the first such studies to use individuals as treatment groups, no feasibility study, pilot, or effect estimate was available to inform our study design. To overcome this limitation, we conducted power simulations using pre-baseline outcome data for the trial cohort using a range of plausible effect sizes. These simulations indicated that no one city was likely to have sufficient availability of offenders or frequency of outcomes to permit a well-powered trial within the funding period of the project. Accordingly, the study was set up as a multisite trial to achieve a large enough sample size for a robust experimental design of individual-level interventions. The success of this approach depends on consistent intervention delivery and evaluation across all sites. However, achieving such consistency is challenging in complex community settings with multiple partners. The protocol for this evaluation has been preregistered [18] and it notes that manualised interventions are unlikely to be exactly replicated in different violence prevention contexts. Despite receiving comparable resources and guidance, the five sites designed similar, but not identical, interventions based on their local contexts.

Without a feasibility study for a multisite trial, this study includes careful process monitoring to check treatment fidelity across sites and tries to control intervention and evaluation processes as much as possible. Since the evaluation team is not responsible for the programme design or delivery, ongoing observation and assessment are vital to justify pooling study data. The process evaluation will check the consistency of delivery throughout the trial period.

Our divergence from group to individual treatment units represents a trade-off between the feasibility of achieving sufficient statistical power and the theory that a group-level mechanism is essential for a focused deterrence intervention to be effective. In effect, a randomised controlled trial using a group as the treatment unit is not possible within available resources and England and Wales population parameters (the remit of Youth Endowment Fund). The multisite design will let us assess the between-site variability of an intervention that, on paper, has the same core components. Combining this with a realist evaluation, we can then try to disentangle how implementation has varied. This can, however, be challenging for multi-agency interventions developed within complex environments.

## Intervention sites

This multisite trial, spanning five English cities—Coventry, Leicester, Manchester, Nottingham, and Wolverhampton—implements seven distinct interventions, all structured around the YEF Focused Deterrence framework (see Table 1). While each intervention adheres to the same foundational aims and theoretical underpinnings, adaptations have been made to accommodate variations in local contexts, resource availability, and organisational structures. For a comprehensive understanding of each intervention's unique and common elements, detailed accounts, structured following the TiDieR (Template for Intervention Description and Replication) framework, are available on the Open Science Framework: https://osf.io/pvnj2/.

**Table 1. Seven interventions across five sites and four delivery teams.**

| Trial number | Team | Site | Intervention name |
|---|---|---|---|
| 1 | Leicester, Leicestershire and Rutland Violence Reduction Network | Leicester City | The Phoenix Programme |
| 2 | Greater Manchester Combined Authority | Manchester City | Another Chance Manchester |
| 3 | Nottingham Violence Reduction Unit | Nottingham City | Another Way |
| 4 | West Midlands Police | Coventry City | CIRV Coventry high risk pathway |
| 5 | West Midlands Police | Coventry City | CIRV Coventry referral pathway |
| 6 | West Midlands Police | Wolverhampton City | CIRV Wolverhampton high risk pathway |
| 7 | West Midlands Police | Wolverhampton City | CIRV Wolverhampton referral pathway |

## Eligibility criteria and recruitment

Participants who might be eligible for the intervention are identified through police records and, in exceptional circumstances, may be referred through statutory services. A set of strictly defined eligibility criteria is then used by each site to identify potential participants from this pool. For example, in Nottingham, participants must:

1. live within the defined boundaries for the site area;

2. have group bonds formed from time spent in the area or have familial ties to the area;

3. have been arrested for violence against a person, robbery, or weapon possession in the 12 months preceding the start of the intervention;

4. have been arrested for one or more of the following offences as part of a group of three or more in the preceding 24 months: violence against the person, criminal damage and arson, robbery, drug offences, possession of weapon offences or public disorder.

Multi-agency programme panels then triage this shortlist of eligible participants at each site. The triaged list is also assessed for deconfliction (i.e., the removal or suspension of anyone subject to active police enforcement for organised crime activity) and the eligible cohort is then randomized into treatment and control groups.

## Statistical power

In the absence of a feasibility or pilot study on which to base estimates of statistical power, we sought to simulate the trial and to deduce a range of required sample sizes based on plausible effect sizes.

**Data.** As part of the preparation phase, the five sites were asked to develop selection/eligibility criteria for the intervention, to identify the number of these individuals in their population, and to describe the number of police-recorded violence against the person offences attributed to them in a twelve-month period. This information allowed us to describe the distribution of outcomes (negative binomial) and to identify the anticipated number of individuals in treatment and control groups.

**Effect size.** As the focused deterrence literature is largely based on population-level treatment effects using quasi-experimental designs and a set of outcomes (e.g. firearm homicide) that are very rare in a UK context, the study could not anticipate an effect size using the existing literature. However, other intervention types with a population at high risk for violent offending have used violence outcomes and randomised designs. The most promising intervention in the YEF Toolkit [19] is Cognitive Behavioural Therapy. CBT interventions are associated with a 25% reduction relative to controls [20]. Using this statistic as a guide, we proposed that effects of 10%, 20%, 30% and 40% relative reductions in police-recorded violence against the person would be reasonable effect sizes.

**Simulation.** Based on this information, we created 10,000 simulated data sets with a similar distribution of the outcome for each of 32 combinations of effect size and sample size (see Table 2). Effect size was simulated by artificially manipulating the outcome for the treatment group and this was modelled using a negative binomial regression analysis. For each combination of effect size and sample size, the proportion of the 10,000 estimates of treatment effect that were statistically significant was stored and is visualised in Fig 1. Reproducible code for these analyses is included as a S1 File.

The graph illustrates the relative importance of effect size ('relative change') and sample size to statistical power. Using these simulations, we concluded that it is unlikely that five sufficiently powered trials would be achievable, even in the most optimistic of treatment effects.

**Table 2. Simulated data sets.**

| n | Relative reduction | | | |
|---|---|---|---|---|
| | 0.9 | 0.8 | 0.7 | 0.6 |
| 100 | 0.0154 | 0.042 | 0.2275 | 0.3248 |
| 200 | 0.0231 | 0.091 | 0.4958 | 0.6393 |
| 300 | 0.029 | 0.1349 | 0.6803 | 0.8371 |
| 400 | 0.0354 | 0.1859 | 0.8255 | 0.9391 |
| 600 | 0.0348 | 0.2745 | 0.954 | 0.993 |
| 800 | 0.0397 | 0.3447 | 0.99 | 0.999 |
| 1700 | 0.04 | 0.6461 | 1 | 1 |
| 2500 | 0.0417 | 0.8334 | 1 | 1 |

Consequently, as the site interventions are largely homogenous—designed according to the same framework and with comparable populations—an option to pool the data from all five sites as a multi-centred trial was considered. This pooling would achieve a sample size of approximately 1,700. Based on this sample size, a relative reduction of 26% would be detectable in 80% of trials. Therefore, pooling was determined to be the best trade-off in terms of value for money and feasibility to detect a plausible effect.

## Summative impact evaluation

The summative impact evaluation has three research questions that will be examined during the 12-month follow-up period per participant. For evaluation purposes, the start of the intervention is operationalised as the point of randomization. The end is set as either the graduation from the programme (i.e., the point at which the delivery team and the programme participant jointly agree that the individual is no longer benefitting from the programme) or as

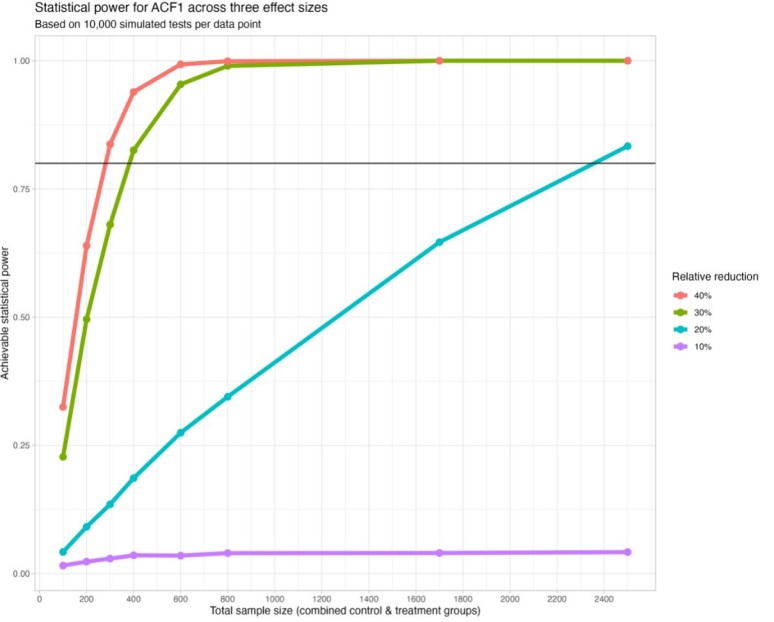

**Fig 1. Statistical power across three effect sizes.**

dropping out from the programme due to, for example, imprisonment or death. Since it is unlikely that each individual will become a part of the programme immediately after being randomized into the treatment group, time from randomization to first contact by the delivery team is recorded in the process data and will be accounted for in the analysis.

The summative impact evaluation questions are as follows:

1. What is the difference in the number of violence against the person offences attributed to individuals receiving the focused deterrence intervention compared to those of similar individuals receiving business-as-usual support?

2. What is the difference in the time to a violence against the person offence (in days) attributed to individuals receiving the intervention compared to those of similar individuals receiving business-as-usual support?

3. What is the difference in the number of co-offending crimes (i.e., crimes involving two or more perpetrators) attributed to individuals receiving the intervention compared to those of similar individuals receiving business-as-usual support?

## Randomisation

Eligible individuals within each site are randomised to treatment or control conditions on a 1:1 allocation ratio. Randomisation is stratified on (i) a three-point ordinal scale of frequency of offending in the past 24 months (derived from raw counts for each location but coarsened for randomisation to low, medium and high) and (ii) whether eligible participants are 18 years old and over or under 18 years of age. Stratification helps to reduce between-group differences at baseline, helping to increase statistical power with a fixed sample size [21].

Prior offending categories (tertiles) have been established from the first cohort of eligible individuals and are site-specific (i.e., grouping into low, medium and high might differ between sites). This method was chosen because it is transferrable across jurisdictions, easily accessible to analysts, and the data-generating process is relatively consistent across areas.

Our second stratification is whether a case is an adult (18 and over) or a child (under 18 years) since all sites expect a mix of such cases (noting that the minimum age is 14 years). This is necessary because statutory services in England generally treat children and adults differently, with deterrence and support provisions varying by child/adult status.

The evaluation team undertook the initial randomisation of long-listed eligible candidates as 'batches'. Further randomisation will be undertaken on a case-by-case basis, as per a 'trickle trial' [22], using a dedicated randomisation platform that will record the individual's unique ID, the site, their tertile of offending frequency, whether they are a child (under 18) at the time of randomisation, and treatment allocation. Records of treatment allocation are accessible to the trial's delivery team and the evaluation team for data retention (trial duration plus 10 years). Random allocation of cases will be completed using the 'randomizr' package [23] for R v4.0.4 [24]. The sample code is recreated and available on the Open Science Framework: https://osf.io/pvnj2/.

## Outcome measures

The primary outcome for the impact evaluation is the number of violence against the person offences attributed to an individual within one year of randomisation. The secondary outcomes are:

1. the number of days between randomisation and a recorded violence against a person offence with a Police National Computer (PNC) disposal outcome relevant to the evaluation;

2. when there is a co-offender, the number of recorded violence against the person offences attributed to an individual with a PNC disposal outcome relevant to the evaluation; and

3. when there is a co-offender, the number of any recorded offences attributed to an individual with a PNC disposal outcome relevant to the evaluation.

For the purposes of this evaluation, PNC disposal outcomes relevant to the evaluation refer to all offences specified by the Home Office's offence codes used in the court proceedings database as Violence Against the Person offences. Police-recorded violence against the person offences measure is the best outcome to evaluate the intervention's effectiveness as it is a standard measure of violent behaviour [25]. Despite about half of all violent behaviour not being reported to the police, this rate decreases as the violence severity increases. Approximately 79% of violence against adults treated by a medical professional is captured, representing the most serious violent offending. However, reporting rates of violence against 10–15-year-olds are considerably lower, with less than 10% of all violence and 18% of violence resulting in medical treatment being reported [26].

These outcome offences will be tracked through an individual's PNC record. PNC metadata will be used to identify cases and outcomes related to this measure. The primary outcome is determined from PNC records using the date of any eligible offence within the trial period. The evaluation team will create a dataset using the date of randomisation as the starting point and count the number of relevant incidents within one year of allocation. This approach was chosen to measure the primary outcome and address some of the known inconsistencies in the crime recording system and Home Office outcomes codes. Primary outcome data will be collected throughout the trial and analysed during the evaluation phase. Access to data and the secure transfer and processing of the data, as well as safe reporting of results, will be governed by data processing agreements between the research team and local police forces.

## Analysis

Statistical significance tests will not be carried out to assess baseline balance, as their premise does not hold in randomised controlled trials (i.e., given that appropriate randomisation procedures were followed, any differences between control and treatment groups at baseline will be due to chance. See http://www.consort-statement.org/checklists/view/32-consort/510-baseline-data) [27]. Instead, tables of the pooled means (and standard deviations, where appropriate) for each characteristic and the magnitude of any differences explored will be presented. For skewed variables, quartile-based measures will be presented. In the statistical analysis plan, we will specify the details for assessing imbalance, which will set out criteria against variables used in randomisation and any putative time control variables used in our analysis to increase power (e.g., previous offending). We will also present balance visually—so, for previous offending, we will look at the distribution of offence counts by treatment and control.

**Intention to treat analysis.**   Based on the project's progress to date and our understanding of the proposed implementation, our analytical approach will pool individual data from all sites into a single analytical model. Our primary analysis will be based on intention-to-treat (ITT). That is, individuals will be analysed according to the group they are randomised to, regardless of whether they engaged with the intervention or remained in control. The ITT approach is particularly relevant for future policy-making stakeholders and practitioners who may roll out or implement a particular intervention without much control regarding how that

intervention is taken up in the system. Therefore, the ITT approach allows for estimating the effects of offering that intervention and incorporates the imperfect uptake of that offer.

The statistical model for the ITT analysis is set out below in Eq 1. The model incorporates variables for between-site and over-time variation and variables used for stratification.

$$Y = \alpha + 1[treatment] + 2[offending\ frequency] + 3[adult] + 4[site] + 5[year/month] + \epsilon\ (1)$$

In the equation, *Y* is the outcome—in this case, a count variable measuring the number of violent offences attributed to an individual in the twelve calendar months following the randomisation of an individual. The analysis approach will be based on count outcomes. We intend to use a *zero-inflated Poisson* (as reflected in our power simulations) because many individuals will likely not have further offences in the follow-up period of 12 months post-randomisation. 1[treatment] will be a binary variable where 0 = control and 1 = treatment, and the coefficient from this variable in the model will be the focal result for the project. 2[offending frequency] is one variable used for stratification in each site. We acknowledge that there is a difference between the model used in the power calculation and the analysis model presented here (S1 File). We will update the power calculations for our statistical analysis plan once we have sample data from sites to understand the likely outcome distribution (based on pre-intervention outcome data). This will be included in the analysis as n-1 dummy categories, with the reference category being the category with the largest number of observations from low, medium, or high offending frequency (note that not pre-specifying which category now will not affect the results). 3[adult] will be a binary variable for whether an individual is an adult (over 18) (= 1) or a child (under 18 years of age) (= 0). 4[site] will be site fixed effects—dummy variables for n-1 sites, again with the site with the largest number of observations overall as the reference category (anticipated to be either Coventry or Wolverhampton in the West Midlands). We include this variable because we know, *a priori*, that sites will differ in their eligibility criteria and selection processes. Hence, we need to account for between-site variation in our analysis. Finally, 5[year/month] will be a variable that captures the year and month since the start of the delivery period. This measure will capture seasonal variation and any esoteric shocks during delivery. This will be entered as a continuous variable, but if there are model convergence problems, we will aggregate this to year-quarter. We also acknowledge that it may be necessary to include another variable for offender sex, depending on the number of females included. If necessary, this would be a binary variable with 0 = male and 1 = female, determined by sex at birth, if possible, to determine this from available data.

For our analysis, we will use robust standard errors and calculate 95% confidence intervals based on those (the exact specification will be clarified in the statistical analysis plan). Standard error adjustment is sensible given that this helps in the event of model misspecification and heterogeneous treatment effects [28, 29]. The model specification will be the same for primary and secondary outcomes.

**Multiple outcome testing.** The study has a single primary outcome, as presented above. That will be the main result reported for the study and will not be adjusted for tests on secondary outcomes. We have secondary outcomes: (i) one on crime and (ii) two on co-offending. The study will adjust the co-offending analysis using the false-discovery rate [30]. All other analyses will be unadjusted. We know, *a priori*, that the study is not powered for sub-group analyses—any such analyses (along with anything not pre-specified) will be clearly labelled as exploratory.

**Blinding during analysis.** While the team are independently appointed evaluators, we will conduct analyses using masked treatment groups, so the analyst conducting the analysis will be blind to allocation.

## Formative process evaluation

The formative process evaluation will use a realist approach to explain how the study worked, in what context and with what population groups. It will address nine formative research questions:

1. To what extent were the three components of the intervention, as required by the YEF framework, received by the treatment population groups?

2. How did inputs contribute to the intervention functioning?

3. Who did the intervention work for, and how?

4. How did local context affect intervention delivery?

5. To what extent was the intervention delivered as intended?

6. How did complexity affect intervention delivery?

7. How did proximal outcomes change?

8. Why did proximal outcomes change?

9. What was learned from how the intervention was delivered?

The formative evaluation uses a mixed methods design, including qualitative, semi-structured cross-sectional and longitudinal interviews, and written researcher notes following the observations of regular meetings between the programme delivery team and programme participants. In addition, it uses quantitative, routine outcome performance monitoring deriving from administrative data. This includes a standardised set of criteria that the sites will collect as a part of the intervention and consists of, for example, the length of time needed for the participant to consent to being in the programme; the number of participants per delivery team member (community navigator/mentor); the number of meetings per participant; the length of being in the programme; the number of referrals to support services; and the number of referrals for enforcement. Sampling for interviews and observations varies, with researchers aiming for more individuals included in the first round of cross-sectional interviews and retaining as many as possible for repeat, longitudinal interviews and observations. Both interview and observation participants will be recruited with support from the delivery team, to reduce the possibility that participating in the study would discourage them from continuing the programme (e.g., out of fear that evaluators work with the police).

In this setting, high-quality logic models and context-mechanism-outcome (C-M-O) configurations are important to ensure the evaluability of the intervention. As shown in Table 3,

**Table 3. Initial high-level C-M-O configurations (v2.0).**

| Context + | Mechanism = | Proximal outcome(s) |
|---|---|---|
| Heterogeneity in local resource availability and deterrence activities | [Targeted enforcement]<br>An increase in legitimate targeted deterrence activities affects perceptions of certainty of arrest and punishment | Normative and instrumental non-violent modifications and engagement with individualised support |
| Variance in resource allocation, coordination, engagement, collaborations and spectrum of available support services | [Individualised support]<br>Increase in availability of improved individualised support packages provides pathways to desistance and increasing perceptions of benefits | Sustained engagement with pathways to desistance and reorientation towards legal and social norms |
| Decrease in levels of community confidence in local policing and statutory and non-statutory support services | [Community validation]<br>Social amplification of community moral voice, characterised by peer and familial influences, informal social controls, collective efficacy, and communicating shared values and beliefs, affects normative compliance and behaviour modification | Increase in legitimacy and support for intervention and enforcement activities to achieve normative compliance |

these initial high-level C-M-O configurations will be used to test and develop a range of realist causal explanations that can be attributed to local 'observable' contexts. As an external expert (Professor Chris Bonnell, London School of Hygiene and Tropical Medicine, UK) recommended, the evaluation contains limited C-M-O configurations centred on generally accepted 'big' ideas relevant to focused deterrence. Those have been previously used in the literature to plausibly explain why targeted deterrence, the provision of support, and community voice and legitimacy may affect specific individuals in varied contexts. During the early implementation phase, a final list of C-M-O configurations will be compiled for evaluation during the full implementation phase. The final list of conditions, aligned to the YEF implementation framework, will likely depend on the availability of relevant data, resources and the viability of testing configurations.

The formative evaluation mainly focusses on targeted deterrence, individualised support, community validation and the interactions between these mechanisms, and the summative outcomes focus on violent offending and involvement in group violence.

## Data collection

The schedule for gathering data began in May 2023 and will run for at least two years. Various methods will be used to gather cross-sectional, longitudinal, and pre-post data from a variety of corroborative data sources. Data from the formative component will be thematically analysed to gain a shared understanding of how, why, and for whom the intervention worked in varied local contexts. The ability to delve deeply into the longitudinal qualitative data collected is a key advantage of the protocol design. Data collection methods include:

**Semi-structured interviews.**   The experiences of programme participants, the delivery team, and stakeholders will be the focus of interviews. Due to the expected high dropout rates among programme participants, we will use convenience sampling and not limit the number of participants. We will offer vouchers equal to the UK living wage (~£15/h) as incentives to retain participants until the end of the intervention.

**Observations.**   A standard observation protocol will be used to monitor participant and delivery team interactions, which will be influenced by the nine YEF FD framework criteria and each location's unique intervention design. Researchers will document regular activities using structured and unstructured methods, noting discussion topics such as group dynamics, decision-making, available resources, and cultural contexts.

**Administrative data.**   The delivery teams are responsible for gathering a wide range of intervention delivery and outcomes data. These data will help characterise the cohort, evaluate the balance between the intervention and control groups, outline participants' journeys through the intervention, measure the degree of engagement, and detail when dropouts and completions occur. This information will be transferred to the evaluation team at regular intervals.

## Discussion

This protocol outlines the process and impact evaluation methodology for the most extensive multisite evaluation of focused deterrence in the UK. Spanning five distinct sites with seven trials, the evaluation includes a cohort of approximately 2,000 individuals, marking it as the only multisite trial of focused deterrence. The protocol uses an integrated realist evaluation framework with a multi-site parallel RCT design to evaluate the efficacy of focused deterrence as an intervention for reducing violent offending among individuals aged 14 and older who are at risk of violent offending in England. To the authors' knowledge, this will be the first multi-site impact and process evaluation of a UK-based violence reduction intervention. The anticipated

findings will offer pivotal insights for formulating future violence prevention policies in the UK. They are also expected to contribute significantly to the corpus of literature on violence prevention and intervention evaluation.

## Interim analysis and 'backfire' check

To mitigate the risk of iatrogenic effects, interim outcome analyses will be conducted roughly six months after the trial start date, defined as the date of first randomisation in any site. These analyses will aim to identify any adverse effects and/or clear evidence of harm, given the offending seriousness of the cohort involved, focusing on the direction and magnitude of the effects. However, it will not involve statistical analysis to avoid 'alpha spending' [31]. In the context of this intervention, harm is defined as a negative impact on offending (averaged across sites) where the reoffending prevalence in the treatment group is equal to or more than 10 percentage points greater than the control group prevalence (e.g., 20% in treatment, 10% in control).

Likewise, to act upon the potential risk of harm, we established a set of pre-specified limits or stopping rules that will inform the decision on the study continuation post the initial six-month period. These stopping rules allow for site-specific differences and avoid terminating the intervention in all sites if one site shows harmful effects during the interim analyses. The rules are:

1. if the average impact is negative, the study will pause intake for one month to allow for a discussion of options and agreement on study progression

2. if the average impact is positive, the threshold for roll-out to all participants will increase. To err on the side of caution, in this case, the reoffending prevalence would need to be 15 percentage points lower in the treatment group than control rather than 10 (e.g., 30% in control, 15% in treatment).

## Ethics and confidentiality

This study has been reviewed and approved by the Faculty of Arts, Cultures and Education Ethics Committee of the University of Hull (ref. 2223STAFF14) with no modifications required. The ethics application included a participant information sheet for each population group (i.e., programme participants, delivery team, stakeholders) and each research method (i.e., interview, observation, survey), a consent form for each population and method, as well as an assent form for parents or carers of those under the age of 18. It further included draft interview schedules, observation themes, survey questions, and risk assessments.

To ensure the confidentiality of all participants, data used in all publications will be anonymised and reported in aggregate, where possible. If direct quotations are used, the researchers will take necessary precautions to ensure that participants cannot be identified. There will be no direct links between the study outcomes (i.e., police-recorded offences) and qualitative data gathered for evaluation purposes.

## Data management

Data management and storage will comply with the UK Data Protection Act 2018 and YEF and University of Hull policies and procedures. To facilitate secure information transfer, we have access to encrypted CJSM accounts, and our team members have NPPV Level 2 security vetting. Data transfer, storage, and processing for summative evaluation will be done through the University of Hull's Data Safe Haven. This system, and protocols for accessing and

processing data, exceed the DSP tools standards required by NHS Digital for processing personal health data and is ISO 27001 certified, making it one of the most secure data facilities available.

## Dissemination and knowledge transfer

Our knowledge transfer strategy extends beyond the final evaluation report. Our plan ensures that our key findings, impact, and process are widely shared and accessible. This involves multiple structured outputs, collectively designed, which will distil and present our insights to diverse audiences. These will be made available in open access academic journals and showcased at global academic and stakeholder conferences, allowing us to disseminate our findings and engage in valuable dialogue with peers and stakeholders. Such discussions often lead to new insights, future collaborations, and opportunities for iterative refinement of our work.

However, given the high vulnerability of the population included in this sample, specifically in the summative evaluation, and bound by the Data Sharing Agreements signed with each of the police forces included, the authors are unable to make the data fully available without restriction. Likewise, the qualitative data gathered is confidential and might include identifiable information. The consent form specifies that all the data will be presented in a pseudonymised and unidentifiable fashion. Thus, we are unable to make the raw data public.

To promote ongoing learning and dissemination of information throughout the project's implementation, we already incorporated various mechanisms to regularly communicate with YEF, project delivery sites, and stakeholders. Some of the current outputs include strategies for formative and summative evaluation at the programme level, theories of change at the programme level, local monitoring and evaluation plans, and system health assessments. By sharing these outputs, we aim to contribute to a broader body of knowledge, enabling other researchers and practitioners to learn from our experiences and potentially apply our insights in their contexts.

## Supporting information

**S1 File. Power calculations.** The is the code to reproduce the power calculations.
(PDF)

## Acknowledgments

We acknowledge colleagues from the Youth Endowment Fund. We also acknowledge colleagues from each of the five sites for their ongoing support and contributions to the design and development of research questions for the process and impact evaluation that are part of this protocol.

## Author Contributions

**Conceptualization:** Tia Simanovic, Paul McFarlane, Iain Brennan, William Graham, Alex Sutherland.

**Data curation:** Tia Simanovic.

**Funding acquisition:** Iain Brennan.

**Methodology:** Tia Simanovic, Paul McFarlane, Iain Brennan, Alex Sutherland.

**Project administration:** Tia Simanovic, Paul McFarlane, Iain Brennan.

**Writing – original draft:** Tia Simanovic, Paul McFarlane, Iain Brennan, Alex Sutherland.

**Writing – review & editing:** Tia Simanovic, Paul McFarlane, Iain Brennan, William Graham, Alex Sutherland.

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
