## [Decision Letter · Decision Letter 0]

21 Dec 2023

PONE-D-23-36345Focussed deterrence: A protocol for a realist multisite randomised controlled trial for evaluating a violence prevention intervention in the UKPLOS ONE

Dear Dr. McFarlane,

Thank you for submitting your manuscript to PLOS ONE. After careful consideration, we feel that it has merit but does not fully meet PLOS ONE’s publication criteria as it currently stands. Therefore, we invite you to submit a revised version of the manuscript that addresses the points raised during the review process.

We look forward to receiving your revised manuscript.

Kind regards,

Nickolas Zaller

Academic Editor

PLOS ONE

Journal Requirements:

This work was supported by The Youth Endowment Fund under grant number 3928041.

Reviewers' comments:

Reviewer's Responses to Questions

**Comments to the Author**

1. Does the manuscript provide a valid rationale for the proposed study, with clearly identified and justified research questions?

Reviewer #1: Partly

Reviewer #2: Yes

2. Is the protocol technically sound and planned in a manner that will lead to a meaningful outcome and allow testing the stated hypotheses?

Reviewer #1: Partly

Reviewer #2: Yes

3. Is the methodology feasible and described in sufficient detail to allow the work to be replicable?

Reviewer #1: No

Reviewer #2: Yes

4. Have the authors described where all data underlying the findings will be made available when the study is complete?

Reviewer #1: No

Reviewer #2: Yes

5. Is the manuscript presented in an intelligible fashion and written in standard English?

Reviewer #1: Yes

Reviewer #2: Yes

6. Review Comments to the Author

You may also provide optional suggestions and comments to authors that they might find helpful in planning their study.

Reviewer #1: Thank you to the authors for their dedication to the field and development of this very important work. This is a foundation of the implementation of a program, however does lack much of the data needed for rationale in power and statistical analysis.

In the introduction, I would prefer to see more epidemiology focus on actual UK data, as opposed to citing US data. For a program that is based in the UK, it is a challenge to cite this as a public health problem from the US surgeon general. Data from the UK, especially in types of violence would be imperative in laying the foundation in the introduction as to the need for the program in the UK. Policy differences are important to discuss and also raise enough disparities between communities that may impact your background in the justification of the use of the type of the program. This was not addressed here.

For the methods, you discuss the rarity of the violent events, but you do not give any number of events in the region. This would be helpful for understanding the expansiveness of the problem. For this study, even if you do not have any data, you could report out how many violent events occurred in the previous 5 years in each location. This would assist in learning if the sites selected would be able to meet the expected enrollment. This study states it is key on having adequate sample size. There are no sample size or power calculations listed, however this is stated several times. Please include these. You only include the sample size in the abstract, but not anywhere in your manuscript, along with the calculations. Please include these and explain rationale for the calculation. Overall, the explanations tend to not be clear. You are not explaining what you will be analyzing where, and your description of mixed methods evaluation is not clear as well. Maybe a table to supplement your planned analyses, or restructuring the section to have it grouped to better explain you plans.

Discussion: in your discussion was the first time you address sample size within the actual manuscript (outside of the abstract)- however having that data earlier would be important, as well as clarifying goal sample sizes for each site. It would be helpful to have a figure about your timeline which will add more clarity to your proposed study activities. What are your anticipated outcomes? You discuss that you plan to share on global stages however what are your anticipated outcomes and what barriers do you expect to encounter?

Overall the general organization of the manuscript can be improved to allow better understanding of the study and planned statistical approach.

There are no references included -- will need these with choices for sample size calculation, and in your introduction as well.

Reviewer #2: Summative Impact evaluation: What is the intervention date for these research questions? At randomization? Authors should clarify.

Outcome measures: Q2 and Q3 could use some clarification about what "PNC disposal outcome relevant to the evaluation" means. Could the authors provide an example?

7. PLOS authors have the option to publish the peer review history of their article (what does this mean?). If published, this will include your full peer review and any attached files.

Reviewer #1: No

Reviewer #2: No

---

## [Author Response · Author response to Decision Letter 0]

15 Feb 2024

Please see attached response to reviewer comments in revised submission

---

## [Decision Letter · Decision Letter 1]

11 Mar 2024

Focussed deterrence: A protocol for a realist multisite randomised controlled trial for evaluating a violence prevention intervention in the UK

PONE-D-23-36345R1

Dear Dr. McFarlane,

We’re pleased to inform you that your manuscript has been judged scientifically suitable for publication and will be formally accepted for publication once it meets all outstanding technical requirements.

Kind regards,

Nickolas Zaller

Academic Editor

PLOS ONE

Additional Editor Comments (optional):

Reviewers' comments:

Reviewer's Responses to Questions

**Comments to the Author**

1. Does the manuscript provide a valid rationale for the proposed study, with clearly identified and justified research questions?

Reviewer #1: Yes

2. Is the protocol technically sound and planned in a manner that will lead to a meaningful outcome and allow testing the stated hypotheses?

Reviewer #1: Yes

3. Is the methodology feasible and described in sufficient detail to allow the work to be replicable?

Reviewer #1: Yes

4. Have the authors described where all data underlying the findings will be made available when the study is complete?

Reviewer #1: Yes

5. Is the manuscript presented in an intelligible fashion and written in standard English?

Reviewer #1: Yes

6. Review Comments to the Author

You may also provide optional suggestions and comments to authors that they might find helpful in planning their study.

Reviewer #1: The authors have done a nice job with the revisions.

The introduction relates it well to the prospective site for the intervention, and the study design is more clear.

Thank you for the thoughtful revisions

7. PLOS authors have the option to publish the peer review history of their article (what does this mean?). If published, this will include your full peer review and any attached files.

Reviewer #1: No

---

## [Editor Report · Acceptance letter]

18 Mar 2024

PONE-D-23-36345R1 

PLOS ONE

Dear Dr. McFarlane, 

I'm pleased to inform you that your manuscript has been deemed suitable for publication in PLOS ONE. Congratulations! Your manuscript is now being handed over to our production team.

Kind regards, 

on behalf of

Dr. Nickolas Zaller 

%CORR_ED_EDITOR_ROLE%

PLOS ONE